# Lactoferrin Binds through Its N-Terminus to the Receptor-Binding Domain of the SARS-CoV-2 Spike Protein

**DOI:** 10.3390/ph17081021

**Published:** 2024-08-04

**Authors:** Patrik Babulic, Ondrej Cehlar, Gabriela Ondrovičová, Tetiana Moskalets, Rostislav Skrabana, Vladimir Leksa

**Affiliations:** 1Laboratory of Molecular Immunology, Institute of Molecular Biology, Slovak Academy of Sciences, 845 51 Bratislava, Slovakia; patrik.babulic@savba.sk (P.B.); gabriela.ondrovicova@savba.sk (G.O.); shunyascience@ukr.net (T.M.); 2Department of Genetics, Faculty of Natural Sciences, Comenius University, 842 15 Bratislava, Slovakia; 3Laboratory of Structural Biology of Neurodegeneration, Institute of Neuroimmunology, Slovak Academy of Sciences, 845 10 Bratislava, Slovakia; ondrej.cehlar@savba.sk

**Keywords:** SARS-CoV-2, spike, lactoferrin

## Abstract

Since Coronavirus disease 2019 (COVID-19) still presents a considerable threat, it is beneficial to provide therapeutic supplements against it. In this respect, glycoprotein lactoferrin (LF) and lactoferricin (LFC), a natural bioactive peptide yielded upon digestion from the N-terminus of LF, are of utmost interest, since both have been shown to reduce infections of severe acute respiratory syndrome coronavirus-2 (SARS-CoV-2), the virus responsible for COVID-19, in particular via blockade of the virus priming and binding. Here, we, by means of biochemical and biophysical methods, reveal that LF directly binds to the S-protein of SARS-CoV-2. We determined thermodynamic and kinetic characteristics of the complex formation and mapped the mutual binding sites involved in this interaction, namely the N-terminal region of LF and the receptor-binding domain of the S-protein (RBD). These results may not only explain many of the observed protective effects of LF and LFC in SARS-CoV-2 infection but may also be instrumental in proposing potent and cost-effective supplemental tools in the management of COVID-19.

## 1. Introduction

Currently, Coronavirus disease 2019 (COVID-19) still persists and brings considerable risk of serious health complications, such as long COVID [1]. Thus, it is still necessary to provide supplemental pharmacological tools to mitigate, cure, or even prevent the threat. In this respect, lactoferrin (LF) and its derivative lactoferricin (LFC) have been gaining growing attention. It was shown that both might inhibit infection of severe acute respiratory syndrome coronavirus-2 (SARS-CoV-2), the virus responsible for COVID-19 [2]. The main cell gate for SARS-CoV-2 is the angiotensin converting enzyme-2 (ACE2), to which the virus binds through its spike protein (S-protein), in particular through the receptor-binding domain (RBD) within the N-terminal subunit (S1) of the S-protein [3,4]. Upon attachment to a target cell, SARS-CoV-2 is primed, i.e., the S-protein is proteolytically processed, mainly by the host transmembrane protease serine 2 (TMPRSS2, transmembrane protease, serine). The priming is pivotal for fusion of viral and cellular membranes and for a virus entry to the host cell, eventually [5,6,7]. Once inside the cell, viral RNA is replicated and packed into new virions [8]. According to an increasing number of studies, LF and LFC may interfere with SARS-CoV-2 in several steps in the frame of this pathway [2].

The human glycoprotein LF (hLF, also termed lactotransferrin), a member of the transferrin family, is present in human milk and other body fluids, but also in secondary granules of neutrophils [9,10,11]. LF is a multifunctional protein endowing antibacterial, antifungal, antiviral, antiparasitic, antioxidant, antitumor, anti-inflammatory, and immunomodulatory activities [12,13,14], which primarily depend either on the ability of LF to sequester iron ions or on the potent binding capacity of the positively charged N-terminal region of LF [14,15]. The natural bioactive peptide LFC, yielded upon digestion by pepsin, is derived right from this region. Free LFC retains some of the biological activities of intact LF, but via its peculiar structural properties, it may also convey additional functions [16]. In regard to SARS-CoV-2, it has been suggested that LF and/or LFC might directly block virus infection through hindering the interaction between the S-protein and heparan sulfate proteoglycans (HSPGs) on the membrane of target cells [17,18], through inhibiting virus priming [19,20], or through hampering RNA replication [21,22].

Previously, we found that the synthetic peptide pLF1 derived from the positively charged N-terminus of human LF, encompassed within the natural peptide LFC, inhibited proteolytic activity of the serine proteases, plasmin, elastase, and TMPRSS2. The full-length LF did not display a similar inhibitory capacity [20]. We attribute this difference to the peculiar conformation of free LFC which is distinct to the structure of the N-terminal part when encompassed within the whole molecule [2]. Nevertheless, both the N-terminal LFC and the full-length LF were capable of reducing SARS-CoV-2 infection on the target cells by about 50% [20]. We concluded that this discrepancy might have been caused by multiple blocking effects of LF and LFC on SARS-CoV-2 infection, i.e., not only via the prevention of virus priming due to TMPRSS2 inhibition by LFC, but also through other mechanisms conveyed by LF. It was suggested by others that the latter might hamper virus entry into a cell by blocking the interaction between the S-protein and HSPG, an alternative receptor for SARS-CoV-2, to which the virus also binds via the RBD of the S-protein [18,23].

Here, we reveal, by means of biochemical and biophysical methods, that LF directly binds to the S-protein of SARS-CoV-2, and we determine the thermodynamic and kinetic characteristics of the complex formation. By using synthetic peptides, we have mapped the mutual binding sites involved in this interaction to the N-terminal region of LF, where LFC is derived from, and to the RBD of the S-protein. These results may not only explain many of the observed protective effects of LF and LFC in SARS-CoV-2 infection but they may also be instrumental in proposing potent and cost-effective supplemental tools in the management of COVID-19.

## 2. Results

To test the influence LF might exert on attaching the viral S-protein to host cells, we performed a series of in vitro binding assays wherein we studied the capability of human and bovine LF (hLF and bLF, respectively) to interfere with S-protein binding to its receptors, ACE2 and HSPG. Since the RBD-mediated binding of the S-protein to soluble human serum albumin (HSA) was observed too [24,25], we also included HSA in the experiments. In particular, purified ACE2, HSPG, and HSA were coated onto wells of a plastic plate and incubated with a purified S-protein in the presence of various concentrations of hLF and bLF. Casein was used as a negative control and for non-specific interaction blocking. After co-incubation and washing, the bound material was collected and analyzed by immunoblotting using the anti-S-protein Ab. In all three coatings, i.e., ACE2 (Figure 1A,D), HSPG (Figure 1B,D), and HSA (Figure 1C,D), we detected a significant concentration-dependent inhibition of S-protein binding with both hLF and bLF, most markedly on ACE2. Such a common inhibitory effect on three structurally diverse S-protein binders drew our attention. We hypothesized that this effect might be explained if LF bound to the S-protein directly and, hence, hindered it from binding to various target molecules.

To test this hypothesis, we performed in vitro binding experiments directly with LF and the S-protein. To map potential binding sites within LF, we applied synthetic LF-derived peptides: pLF1 from the N-terminal region (encompassed within LFC), pLF3 from the C-terminal region, and pLF2 from the helix-linker region [26]. Additionally, we used a scrambled peptide pCTR as a negative control. When we incubated the S-protein on the coated LF in the presence of the peptides, we observed about 60% of the input S-protein bound to LF. We detected a significant concentration-dependent reduction in S-protein binding with the N-terminal peptide pLF1, and a partial, not significant, reduction with the C-terminal peptide pLF3. We did not observe any effect with the helix-linker peptide pLF2 (Figure 2A,B). Next, we wanted to examine if the S-protein did bind to LF through the RBD. Thus, we repeated the in vitro binding assay with a purified RBD. In comparison to the full-length S-protein, the binding of the RBD to LF was weaker, i.e., about 30% of the input RBD protein (Figure 2C,D). However, we again observed the concentration-dependent inhibitory effects of pLF1 and pLF3, which were both significant. We did not observe any effect with the helix-linker peptide pLF2 (Figure 2C,D). These findings suggest that LF binds to the RBD of the SARS-CoV-2 S-protein through its N-terminal, the LFC-encompassing region.

Next, to quantify the strength of the interaction between LF and the viral proteins, we used surface plasmon resonance (SPR) to evaluate LF’s binding to both the S-protein and RBD. We focused on hLF in these measurements. hLF was immobilized on the surface of a sensor chip, and the interaction with variable concentrations of viral proteins was monitored using a Biacore 3000 instrument (Figure 3A,B). The interaction with both proteins could be fitted to a 1:1 reaction mechanism. For RBD, the interpolated binding kinetics point to the relatively fast association and dissociation rates, with the equilibrium dissociation constant equal to 1.3 µmol/L. This result was also confirmed by fitting the concentration dependence of RBD steady-state binding levels (Figure 3B, Table 1). The interaction with the S-protein exhibits similar kinetics of association but, contrary to the RBD, more than 400-times slower dissociation, with the dissociation constant equal to 4.3 nmol/L (Figure 3A, Table 1). Data from SPR pointing to the stronger binding of LF to the S-protein, when compared to the RBD, were in agreement with the results obtained from in vitro binding assays (Figure 2). Interestingly, as observed from the level of the SPR signal and considering the molecular weight of both viral proteins, the S-protein apparently bound to LF to lower extent than the RBD during the time used for complex formation (90 s). Such kinetic observations may be related to a trimeric state of the S-protein. It is known that the RBD has different accessibility in an open and closed state of the S-protein trimer [27]. In summary, the S-protein binding experiments suggest that firstly, LF may bind only a minor population of S-protein conformational states with accessible LF binding site(s) within the RBD; secondly, once bound to the chip, S-protein dissociation is slowed down either by an avidity effect (serial binding of the same S-protein molecule by adjacent RBDs) or by a conformational trapping of the LF-bound S-protein.

Finaly, we tested the LF and S-protein interaction in solution. We used the RBD for further experiments. We incubated LF and RBD individually and together, and then analyzed the protein samples by means of two-dimensional blue native electrophoresis (BN-PAGE). After separation and visualization of both proteins individually, we observed the major bands corresponding apparently to monomeric forms of LF and RBD and minor bands corresponding probably to dimeric forms (Figure 3C). When the LF-RBD pre-incubated sample was analyzed, two vertically aligned bands of LF and the RBD corresponding to their mutual complex became apparent in the range about 120 kD, confirming the 1:1 stoichiometry (Figure 3C).

Taken together, we demonstrate here that LF directly binds the RBD of the S-protein of SARS-CoV-2. This interaction may be implicated in the multiple protecting effects of LF in SARS-CoV-2 infection: LF might not only block the binding of SARS-CoV-2 to HSPG but also to its primary receptor ACE2.

## 3. Discussion

Both hLF and bLF together with respective LFCs have been demonstrated to block cell entry of many viruses [2]. It has been shown that LF binds on host cells’ receptors for viruses, such as herpes simplex virus 1 and 2 (HSV-1, HSV-2) [28,29], human immunodeficiency virus 1 (HIV-1) [30,31,32,33,34], dengue virus [35], or coronaviruses [17,18,23]. Specifically, in SARS-CoV-2, LF has been reported to block the interaction between the viral S-protein and the membrane of target cells by binding to HSPG, an alternative viral receptor [17,18]. 

With about 700 amino acids and a molecular weight of ∼80 kDa, hLF is folded into two homologous lobes [36]. Since LF belongs to the transferrin (TF) family, it shares a similar structure with TF. However, the highly positively charged N-terminal region is peculiar for LF. Thus, LF is the most alkaline member of the transferrin family and binds to many negatively charged surfaces [10,11,14,37,38,39]. We have mapped the binding site to the S-protein into this N-terminal part of LF. The synthetic peptide pLF1 derived from the N-terminal region, where the bioactive peptide LFC is derived from, blocked the binding. The highly positively charged LFC is the bioactive peptide released from LF by pepsin cleavage in the gut; thus, the antimicrobial activity of intact LF is commonly attributed to its iron-binding capacity and to specific binding capacities of its N-terminus [2,40].

Nevertheless, structural studies have revealed that in contrast to intact LF wherein the N-terminus adopts a β-α-β-α motif [10], the conformation of the free N-terminal LFC is radically different [2,40,41]. This peculiar structure, resulting from both the high net positive charge and the position of the cationic residues, appeared to also be important for the net effectivity of LF, LFC, or synthetic LF-derived peptides, in the S-protein of SARS-CoV-2 blockade from binding to target cells. Thus, although LFs from different species exhibit high homology [10,14,15,42], they may differ in their antiviral potencies dependent on slight differences in the tertiary structure and charge of their respective N-termini, e.g., bLFC is considered more potent than the human counterpart [40,43,44]. In addition, most studies have examined the antiviral properties of bLFC and hLFC, yet a limited number of studies have also investigated LFC from other species, such as pig, mouse, goat, and camel [43,45,46,47,48,49]. Future in silico approaches, i.e., molecular docking simulations, may be instrumental in identifying the variants of LFC with the uppermost affinity to S-protein and thus highest effectivity against SARS-CoV-2. 

Interestingly, the C-terminal peptide pLF3 also partially blocked the interaction, indicating that both ends of LF take part in the interaction.

Notably, it was observed that the inhibitory effect of LF on SARS-CoV-2 infection was dependent on the iron-saturation state of LF [50]. However, since LFC, pLF1, and pLF3 lack iron-binding capacity [16], we suppose that the iron saturation would not directly influence the binding of LF to the S-protein and that the intact LF also blocks the infection via other mechanisms [2].

Direct binding between LF and the S-protein of SARS-CoV-2 was already suggested; in a pull-down approach, Cutone et al. identified LF to bind various S-protein variants, and through in silico molecular docking simulations, they proposed the model structure for this binding [51,52]. In the model, the RBD was directly implicated in the binding, in agreement with our experimental mapping studies. In the model, the tips of both the N- and C-lobes of LF appeared to be involved in contact with the S-protein, not encompassed within LFC. However, for these in silico studies, structures of non-glycosylated forms of LF were applied, which makes the suggested contact sites questionable since the corresponding parts of both hLF and bLF contain glycosylation sites (N137 and N368, respectively) for relatively long sugar chains.

## 4. Materials and Methods

### 4.1. Materials

Ammonium persulfate, TEMED, SDS, acrylamide, and N,N‘-methylenebis-acrylamide were purchased from SERVA (Heidelberg, Germany). ACE2, heparan sulfate proteoglycans (HSPG), human serum albumin (HSA), casein, both hLF (#L1294) and bLF (#L9507), and all protease inhibitors were from Sigma-Aldrich (St. Louis, MO, USA). The SARS-CoV-2 S-protein was from NativeAntigen (#REC31868, Kidlington, UK) or generated by us as described [20]. The RBD of the S-protein was expressed in the CHO cell line and affinity purified, as described previously [53]. All S-proteins and RBDs were derived from the ancestral Wuhan-like variant. The peptides derived from hLF [20,26] were synthesized by Peptide 2.0 (Chantilly, VA). The sequences of the 19-residue synthetic peptides were as follows: GRRRSVQWCAVSQPEATKC (pLF1; N-terminal residues 1-19), EDAIWNLLRQAQEKFGKDK (pLF2; middle helix-linker residues 264-282), NLKKCSTSPLLEACEFLRK (pLF3; C-terminal residues 673-691), and NFRTKSCPLELAKELKLCS (pCTR). mAbs to human LF (LF5-1D2, LF65-3D5, LF95-4C5, and LF124-5E2) were generated in the laboratory of Otto Majdic. Biotinylated rabbit polyclonal Ab to LF (A53619) was purchased from Abcam (Cambridge, UK), and the mAb to the S-protein was from Sino Biological Co., Ltd. (Beijing, China, #40591-MM42). The HRP-conjugated goat anti-mouse IgG secondary antibody (Pierce Goat Anti-Mouse IgG, (H+L), Peroxidase Conjugated) and the streptavidin–HRP conjugate (RABHRP3) was from Sigma-Aldrich.

### 4.2. In Vitro Binding Assay and Immunoblotting

For the binding assay, various molecules (ACE2, HSPG, HSA, hLF, casein) solubilized in PBS (pH 8.7) at a concentration of 5 µg/mL were coated on wells of a 96-well TPP plate (#92696, Sigma) for 2 h at 37 °C. Then, the wells were blocked with 1% casein in PBS for 1 h at room temperature and washed two times with PBS. Afterwards, the wells were incubated overnight at 4 °C in binding mixture supplemented with the purified assayed proteins—the S-protein or RBD (both 10 µg/mL)—in the absence or presence of the peptides—pLF1, pLF3, pLF2, and pCTR—at increasing concentrations (from 5 to 20 µg/mL). Afterwards, the wells were washed by slowly and continuously immersing the plate into a tray filled with PBS. The PBS was then gently removed and the bound material was analyzed by SDS-PAGE and immunoblotting; in particular, the washed and dripped wells were filled with the SDS-PAGE sample buffer (10 µL/well) and four wells were merged to one sample for the SDS-PAGE. Then, the samples gained from in vitro binding assays were analyzed by electrophoresis on an appropriate SDS–polyacrylamide gel (SDS-PAGE) followed by transfer at constant voltage (15 V) to an Immobilon polyvinylidene difluoride membrane (Millipore Co., Bedford, MA, USA). The membranes were blocked using 4% non-fat milk and immunostained with an appropriate mAb followed by a corresponding secondary HRP conjugate. For visualization of proteins, the chemiluminescence image analyser Azure 280 (AzureBiosystems, Dublin, CA, USA) was used. Densitometric quantifications were performed by means of the AzureSpot software (Version 2.1.097) as follows: The values corresponding to the initial binding mixture (input; both 10 µg/mL) were set as 100%. From the values of the respective bound protein bands, the values corresponding to the casein binding were subtracted as negative controls, and the resultant values of the binding were expressed as percentages of the inputs.

### 4.3. Surface Plasmon Resonance (SPR)

A Biacore 3000 instrument with a streptavidin Sensor Chip SA (GE Healthcare, Chicago, IL, USA) was used. All experiments were performed at 25 °C in PBS (10 mmol/L phosphate buffer, pH 7.2, 140 mmol/L NaCl) with 0.005% Tween 20 as running buffer. Immobilization of hLF on two independent flow cells of the chip was performed as described previously [26]. S-proteins (0.12 µmol/L) or serially diluted RBDs (0.4–10 µmol/L) were injected in duplicates at a 100 µL/min flow rate, allowing an association and dissociation time of 90 and 600 s, respectively. Regeneration of the LF surface was accomplished by two 3 s injections of 10 mmol/L HCl. Binding data were double-referenced [54], and the kinetic constants and affinity were derived from fitting a 1:1 Langmuir model as implemented in BIAevaluation software version 4.1.1. Rate constants and maximal responses were approximated globally, and the bulk response was set to zero. RBD binding was also independently fitted to a steady-state equilibrium model.

### 4.4. In-Solution Binding and Blue Native Polyacrylamide Gel Electrophoresis (BN-PAGE)

The studied proteins, i.e., LF and RBD, were incubated in binding buffer (10 mmol/L Tris, 50 mmol/L NaCl, pH 7.2) at concentrations of 0.1-0.2 µg/µL, separately or together in a ratio of RBD/LF of 2:1 to allow the formation of the complex. Afterwards, the samples were adjusted with the BN-PAGE sample buffer and analyzed by BN-PAGE, as described in detail elsewhere [55]. The protein samples, LF, RBD, LF-RBD complex, and BSA (Sigma-Aldrich, St. Louis, MO, USA) as a marker were loaded on the first dimension native separation gel. Electrophoresis was performed at 80–180 V at 4 °C. Then, the vertical lanes were cut from the gel, put on top of second-dimension SDS–polyacrylamide gel, and run at room temperature. Afterwards, the gels were blotted onto Immobilon polyvinylidene difluoride membranes; the membranes were blocked with 4% milk and then incubated with specific antibodies for visualization.

### 4.5. Statistical Analysis

All microplate experiments were performed at least three times in at least duplicates. Interpolated SPR parameters were averaged from two separated flow cells. The data were expressed as mean values with standard deviations. Statistical significance was evaluated by using Student’s *t*-test or one-way ANOVA with Tukey’s post-test by using the Prism 10 software; values of *p* * < 0.05, *p* ** < 0.005, *p* *** < 0.0005 (as indicated) were considered to be significant or highly significant, respectively.

## 5. Conclusions

The interaction of SARS-CoV-2 with cellular receptors, either ACE2 or HSPG, is a central step involved in the pathogenesis of COVID-19. The outcomes of biochemical and biophysical experiments, presented in this study, provide novel knowledge on molecular determinants of the complex between LF and the S-protein of SARS-CoV-2, which may be instrumental to produce novel supplemental tools of higher effectivity for the treatment of COVID-19.

## Figures and Tables

**Figure 1 pharmaceuticals-17-01021-f001:**
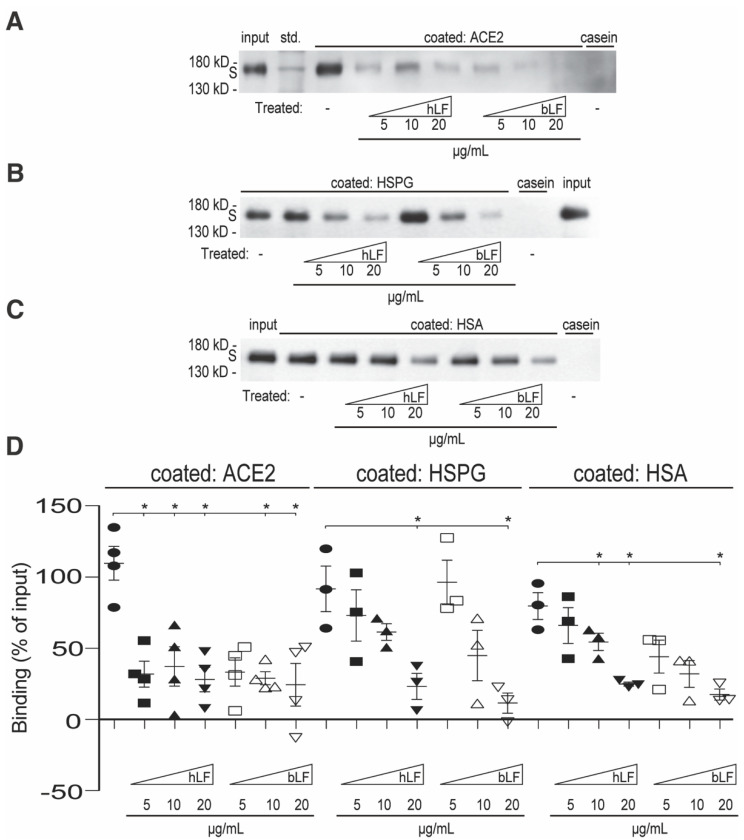
Effect of hLF and bLF on S-protein binding. ACE2 (**A**), HSPG (**B**), and HAS (**C**) were coated on wells of a 96-well plastic plate. Casein was used as a negative control. After blocking, the wells were incubated overnight at 4 °C with the S-protein in the absence or presence of hLF or bLF at indicating concentrations (from 5 to 20 µg/mL). Afterwards, the wells were washed and the bound material was analyzed by immunoblotting using mAb to the S-protein. (**D**) Densitometric evaluation of at least three independent experiments. The values corresponding to the binding mixture (*input*) were set as 100%. The values corresponding to the casein binding were subtracted from the values of the respective bound protein bands; the resultant values are expressed as percentages of the inputs: std., molecular weight standard. The data from at least three blots were evaluated. Values of *p* * < 0.05 (as indicated) were considered to be significant or highly significant, respectively; *std.* corresponds to the lane with the molecular weight standards.

**Figure 2 pharmaceuticals-17-01021-f002:**
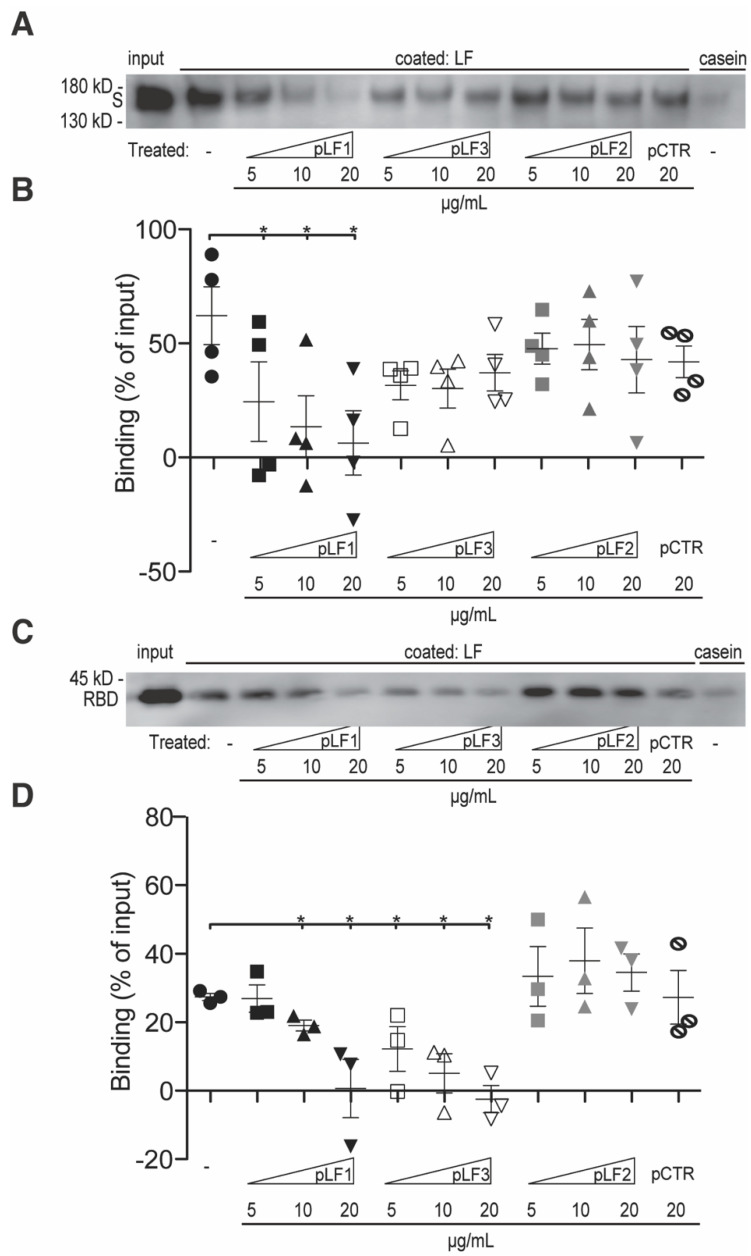
Mapping of the S-protein binding to LF. hLF was coated on wells of a 96-well plastic plate. Casein was used as a negative control. After blocking, the wells were incubated overnight at 4 °C with the S-protein (**A**) or RBD (**C**) in the absence or presence of the peptides—pLF1, pLF3, pLF2, and pCTR, at indicated concentrations (from 5 to 20 µg/mL). Afterwards, the wells were washed and the bound material was analyzed by immunoblotting using mAb to S-protein. (**B**,**D**) Densitometric evaluations of at least three independent experiments. The values corresponding to the binding mixture (*input*) were set as 100%. The values corresponding to the casein binding were subtracted from the values of the respective bound protein bands; the resultant values are expressed as percentages of the inputs. The data from at least three blots were evaluated; values of *p* * < 0.05 (as indicated) were considered to be significant or highly significant, respectively.

**Figure 3 pharmaceuticals-17-01021-f003:**
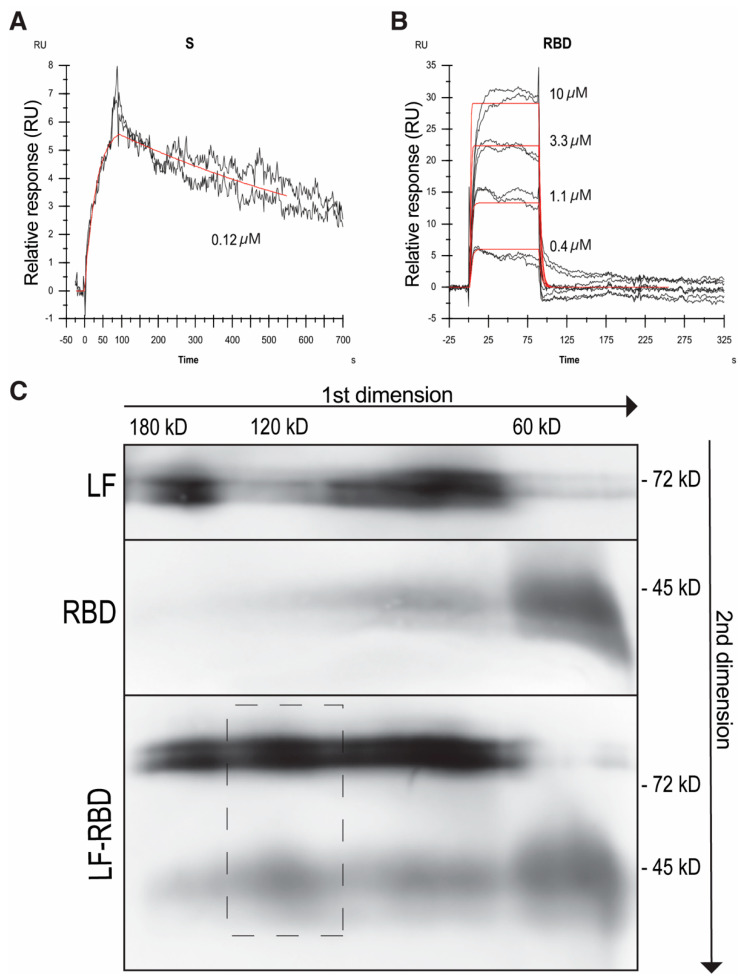
Characterization of the S-protein/RBD–LF interaction. SPR analysis of LF-S-protein (**A**) and LF-RBD (**B**) binding. Duplicates of serially diluted S-protein or RBD at the indicated concentrations were injected over immobilized LF, and SPR response (black) was fitted to 1:1 Langmuir model (red). (**C**) To analyze the LF-RBD complex in solution, samples of LF and RBD were incubated separately and together and subjected to analysis by BN-PAGE in the first dimension followed by SDS-PAGE in the second dimension and visualized by immunoblotting using mAb to LF and S-protein followed by an appropriate HRP conjugate. A dashed rectangle indicates the LF–S-protein complex of about 120 kD size.

**Table 1 pharmaceuticals-17-01021-t001:** Kinetics and affinity of hLF–S-protein binding (average value ± SD, *n* = 2).

	k_a_ [M^−1^s^−1^]	k_d_ [s^−1^]	K_D_ [M]	K_D_ * [M]
LF–RBD	(2.45 ± 0.45) × 10^5^	(3.08 ± 0.85) × 10^−1^	(1.26 ± 0.42) × 10^−6^	(1.31 ± 0.49) × 10^−6^
LF–S-protein	(1.60 ± 1.23) × 10^5^	(6.93 ± 5.76) × 10^−4^	(4.32 ± 4.88) × 10^−9^	N.D.

* Steady-state equilibrium model; k_a_, association rate constant; k_d_, dissociation rate constant; K_D_, equilibrium dissociation constant; N.D., not determined.

## Data Availability

Data is contained within the article.

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
