# Peer review of "Lactoferrin Binds through Its N-Terminus to the Receptor-Binding Domain of the SARS-CoV-2 Spike Protein"

_pharmaceuticals, 2024, doi:10.3390/ph17081021_

Round 1

Reviewer 1 Report

Comments and Suggestions for Authors

Based on immunoblotting and surface plasmon resonance approaches, Babulic and colleagues explored the binding of lactoferrin (LF) to the spike (S) protein of SARS-CoV-2, mapping the mutual binding sites involved in this interaction. The study is well designed and additive to the current scientific literature on potential anti-SARS-CoV-2 strategies. However, some points requires revision and are listed below in the order of appearance in the text:

Lines 73-86: Move the first paragraph of the section 2 (results and discussion) to the section 1 (introduction) to better accommodate its content.

Lines 102-103: Explain why bLF was excluded from the in vitro binding experiments directly with LF and SARS-CoV-2 S protein.

Lines 108-110/113-114: Describe the findings for the helix-linker peptide pLF2.

Lines 115-116: If the C-terminal peptide pLF3 also significantly inhibited RBD binding to LF, it should be suggested that LF binds to the RBD of the SARS-CoV-2 S protein through its C-terminal too.

Lines 117/237: Clarify which LF (human or bovine) was used in the surface plasmon resonance experiment and explain why one of them was excluded from this analysis.

Line 138: When discussing the possible binding mechanism between LF and SARS-CoV-2 S protein, consider the potential role of the iron-saturation state (apo or holo) of the former and the impact of the current omicron variants of the latter in this process (as a suggestion, check out the work by Alves et al., 2023 - Pharmaceuticals 16: 1352 - DOI: 10.3390/ph16101352).

Lines 161/171: Provide the meanings of the asterisks shown in Figures 1D, 2B and 2D in their respective captions, even though they are already explained in the section 4 (materials and methods); moreover, clarify the superimposed asterisks on the binding values for the pCTR groups in Figures 2B and 2D.

Lines 162-163/172-173: Delete the repeated "were coated" in the following sentences: "ACE2 (A), HSPG (B), HAS (C) were coated were coated on wells of a 96-well plastic plate" and "hLF was coated were coated on wells of a 96-well plastic plate".

Line 181: Provide the y-axis title in Figure 3A.

Lines 196-197: Indicate the SARS-CoV-2 variant (ancestral, alpha, beta, gamma, delta or omicron) from which the S protein and its isolated RBD were derived.

Lines 265-267/309-311/319-325/326-328: Remove highlighting from the references 2, 20, 23 and 24.

Comments on the Quality of English Language

Minor editing of English language required.

Author Response

Response to reviewer #1:

We thank the reviewer for the positive evaluation of our manuscript.

Lines 73-86: Move the first paragraph of the section 2 (results and discussion) to the section 1 (introduction) to better accommodate its content.

We thank the reviewer for this notion and moved the text to Introduction [lines 54-63].

Lines 102-103: Explain why bLF was excluded from the in vitro binding experiments directly with LF and SARS-CoV-2 S protein.

We used bLF in the Figure 1 experiments to verify whether also LF from non-human species would block the interaction. The quantitative SPR binding experiments with human protein presented here are a continuation of our previous work [1], where we show the inhibitory effect of hLF on the SARS-CoV-2 infection on cells. Yet, a limited number of studies have also investigated LFC from other species, such as pig, mouse, goat and camel [43, 45-49]. On our follow-up work, we plan to examine the blocking and antiviral properties of not only hLF and bLF, but also from the other species, such as pig, mouse, goat and camel. In particular, we plan in silico approaches, i.e. molecular docking stimulations, which will be instrumental in identifying the variants of LFC with the uppermost affinity to S-protein and thus highest effectivity against SARS-CoV-2. Our next plans are mentioned in the Discussion section of revised manuscript [lines 146, 148].

Lines 108-110/113-114: Describe the findings for the helix-linker peptide pLF2.

We included the requested notion in the revised version [lines 89, 93].

Lines 115-116: If the C-terminal peptide pLF3 also significantly inhibited RBD binding to LF, it should be suggested that LF binds to the RBD of the SARS-CoV-2 S protein through its C-terminal too.

We agree with this notion and we comment this finding in Discussion [lines 149-150]. We plan in our future work to clarify this issue by both in silico and crystallization studies.

Lines 117/237: Clarify which LF (human or bovine) was used in the surface plasmon resonance experiment and explain why one of them was excluded from this analysis.

This information was listed in Methods. Now we put it also in Results. We used hLF for the reasons explained above and consider the usage of LF from other species in the continuation of our research.

Line 138: When discussing the possible binding mechanism between LF and SARS-CoV-2 S protein, consider the potential role of the iron-saturation state (apo or holo) of the former and the impact of the current omicron variants of the latter in this process (as a suggestion, check out the work by Alves et al., 2023 - Pharmaceuticals 16: 1352 - DOI: 10.3390/ph16101352).

We thank the reviewer for this important observation; accordingly, we added our comment on it together with the reference [lines 151-154]. According to the manufacturer, the LF we applied was saturated >90%.

Lines 161/171: Provide the meanings of the asterisks shown in Figures 1D, 2B and 2D in their respective captions, even though they are already explained in the section 4 (materials and methods); moreover, clarify the superimposed asterisks on the binding values for the pCTR groups in Figures 2B and 2D.

The requested information is now involved also in the figure legends. In Figures 2B and 2D, those were symbols corresponding to control values. For better clarity, in the revised form we exchanged the symbols.

Lines 162-163/172-173: Delete the repeated "were coated" in the following sentences: "ACE2 (A), HSPG (B), HAS (C) were coated were coated on wells of a 96-well plastic plate" and "hLF was coated were coated on wells of a 96-well plastic plate".

We apologize for this shortcoming. We corrected the issue.

Line 181: Provide the y-axis title in Figure 3A.

We apologize for this shortcoming. We corrected the issue.

Lines 196-197: Indicate the SARS-CoV-2 variant (ancestral, alpha, beta, gamma, delta or omicron) from which the S protein and its isolated RBD were derived.

We thank the reviewer for this important notion; accordingly, we added the required information [line 216]. All S proteins and RBD are derived from the ancestral Wuhan-like variant.

Lines 265-267/309-311/319-325/326-328: Remove highlighting from the references 2, 20, 23 and 24.

We apologize for this shortcoming. We corrected the issue.

Reviewer 2 Report

Comments and Suggestions for Authors

1.     In title of the manuscript, the authors should add not only lactoferrin but also lactoferricin. Furthermore, the title described only the compound bind the S protein but no information after binding it. The authors should revise the title of the manuscript. 

2.      In this study, the authors performed the invitro experiments using S protein. Why did not the author investigate with live virus using cell culture method? This is very important for proving the lactoferrin can reduce the virus entry into the v\cell and prevent the infection.

3.     The authors investigated only the compound binds to S protein and they did not show this binding really prevent the infection by means of blocking the virus entry into the cells. The authors should do invitro and invivo experimens for proving the compound has really antiviral activity.

4.     The authors should add limitations of the study such as the authors investigate the binding of the compound to S protein and not investigating for really preventing the infection or not at the revised manuscript.

Author Response

Response to reviewer #2:

We thank the reviewer for the positive evaluation of our manuscript.

  1. In title of the manuscript, the authors should add not only lactoferrin but also lactoferricin. Furthermore, the title described only the compound bind the S protein but no information after binding it. The authors should revise the title of the manuscript.

Since we did not test LFC in our assays, we would prefer to keep the original title. We suppose that it conveys the most important message of our work – first, LF binds to S protein; and second, what are the molecular determinants – the N-terminus and RBD In LF and S protein, respectively.

  1. In this study, the authors performed the invitro experiments using S protein. Why did not the author investigate with live virus using cell culture method? This is very important for proving the lactoferrin can reduce the virus entry into the v\cell and prevent the infection.

Our focus here was on the mechanism how LF and LFC exert their inhibitory effect on the SARS-CoV-2 infection which we demonstrated before directly on various target cells [1]. The presented manuscript is thus the follow-up of the latter paper. In particular, we found that the synthetic peptide pLF1 derived from the positively charged N-terminus of human LF, encompassed within the natural peptide LFC, inhibited proteolytic activity of the serine proteases, plasmin, elastase and TMPRSS2. The full-length LF did not display similar inhibitory capacity [1]. We attributed this difference to the peculiar conformation of free LFC which is distinct to the structure of the N-terminal part when encompassed within the whole molecule [2]. Nevertheless, both, the N-terminal LFC and the full-length LF, were capable of reducing SARS-CoV-2 infection on the target cells by about 50% [1]. We concluded that this discrepancy might have been caused by multiple blocking effects of LF and LFC on SARS-CoV-2 infection, i.e. not only via the prevention of virus priming due to TMPRSS2 inhibition by LFC, but also through other mechanisms conveyed by LF. It was suggested by others, that the latter might hamper the virus entry into a cell by blocking the interaction between S protein and HSPG, an alternative receptor for SARS-CoV-2, to which the virus also binds via RBD of S-protein [3, 4]. Our presented work aims to clarify this discrepancy. This intention is noted in the revised version in Introduction [lines 54-63].

  1. The authors investigated only the compound binds to S protein and they did not show this binding really prevent the infection by means of blocking the virus entry into the cells. The authors should do invitro and invivo experimens for proving the compound has really antiviral activity.

As described above, the antiviral capacity of the compounds was already demonstrated by us in the cellular models. Here, we indented to contribute by revealing the possible molecular mechanism.

  1. The authors should add limitations of the study such as the authors investigate the binding of the compound to S protein and not investigating for really preventing the infection or not at the revised manuscript.

This is now described in Introduction [lines 54-63].

Reviewer 3 Report

Comments and Suggestions for Authors

For this article “Lactoferrin Binds Through its N-terminus to the Receptor-binding Domain of the SARS-CoV-2 Spike Protein”, I can declare my report of reviewing. The topic is indeed at a high level of importance because of trying to provide new approach for dealing with the coronavirus infections. The authors mentioned the importance of their work regarding the available threats of COVID-19. They tried to explore functions of the glycoprotein lactoferrin (LF) and lactoferricin (LFC) to reduce the impacts of virus infection. In this case, a binding process was happened to manage such infection issue by means of biochemical and biophysical methods. The binding of compounds-virus were found applicable to the S-protein of coronavirus. They claimed that their achievements could be useful for dealing with the infection and also a cost-effective tool could be introduced for the management of COVID-19.

In my opinion, this work could be accepted, but I can give some comments:

1- Please describe more the extraction of lactoferrin and lactoferricin compounds, how did you purify them?

2- Did you check the toxicity of lactoferrin and lactoferricin compounds for the normal cells?

3- Please also arrange the whole Material and Method part in a more details version. 

4- For Figures 1 and 2, is there any time table of obtaining the results?

5- Is there any temperature programming for obtaining the results?

6- For Figure 3, Panel A, should we expect a constant red line in the right side of diagram? It is still descending!

7- Did you study if your binding is reversible or irreversible? In each case, how do you suggest for optimizing the binding process?

8- I can advise you to include these two citations in your work and also make a comparison with your own achievements 

https://doi.org/10.1097/MD.0000000000032971

https://doi.org/10.1002/ddr.21895

Author Response

Response to reviewer #3:

We thank the reviewer for the positive evaluation of our manuscript.

1- Please describe more the extraction of lactoferrin and lactoferricin compounds, how did you purify them?

Both hLF and bLF were purchase in a purified form, the LFC-derived peptide pLF1 and all other peptides were synthesized, all as it is described in Materials and methods. Since the natural LFC was not applied in the presented work, we do not describe its preparation. It is described in detail by us elsewhere [1].

2- Did you check the toxicity of lactoferrin and lactoferricin compounds for the normal cells?

Safety of LF and LFC was tested and confirmed in many independent in vitro and in vivo studies. LF is one of the most abundant glycoproteins in mother milk and it is used as supplement in neonatal medicine. LF and LFC have been also examined in several clinical trials that focused on their potential application in treatment of variety of human diseases. This has been possible because the European Food Safety Authority (EFSA, 2012) [5] and the U.S. Food and Drug Administration (FDA, 2014; GRAS notice, GRN 465) recognized LF as a food supplement and as a part of infant formula as generally safe. Accordingly, we did not notice any toxic effects of LF and LFC in our cellular models.

3- Please also arrange the whole Material and Method part in a more details version.

We provide in the revised version all required details in the Material and method section.

4- For Figures 1 and 2, is there any time table of obtaining the results?

The results were obtained from 2022 to 2024.

5- Is there any temperature programming for obtaining the results?

The temperature conditions are described in the Materials and Method section. For binding, we used 37 °C or 4 °C as indicated.

6- For Figure 3, Panel A, should we expect a constant red line in the right side of diagram? It is still descending!

The shape of the S-protein dissociation phase (right side of Fig.3A – red line) is traced by a least-square interpolation of experimental binding curves – black lines – by the BIAevaluation software. As the reviewer is pointing out, at the end of protein dissociation, when only a negligible amount of protein is still bond to the chip, the interpolated red line should approximate a line parallel to the time axis, with a constant value. However, the time needed for interpolated red line being constant depends on the value of the dissociation rate constant. For RBD (Fig.3B) the dissociation rate constant is very fast (~0.3 s-1, Table 1), therefore constant trace is achieved already after a few seconds of dissociation phase. In contrast, for S-protein in the panel A the dissociation rate constant is nearly 500-times slower (~0.0007 s-1, Table 1), therefore for achieving a constant dissociation trace for S-protein with a complete dissociation we would need to follow the dissociation for hours, which is not needed for the purpose of the experiment. 

7- Did you study if your binding is reversible or irreversible? In each case, how do you suggest for optimizing the binding process?

We did not study the reversibility of binding in a separate experiment, however, deducing from the determined Langmuir 1:1 binding mode on SPR we can expect a reversible binding. Optimization of binding process would be important for increasing hLF inhibition of viral infection and we plan to achieve this aim e.g. by in silico experiments, as discussed in the revised manuscript.

8- I can advise you to include these two citations in your work and also make a comparison with your own achievements

https://doi.org/10.1097/MD.0000000000032971

https://doi.org/10.1002/ddr.21895

We thank the reviewer for the suggestions, nevertheless, since the references do not significantly enhance our manuscript, we have decided not to include them.

Reviewer 4 Report

Comments and Suggestions for Authors

In the present work, Babulic and colleagues, by biochemical and biophysical methods, report a direct binding of lactoferrin to the receptor-binding domain of the SARS-CoV-2 spike protein. Overall, the paper is well written and provides comprehensive information on the subject. However, I have a few observations to make:

-       I suggest moving the abbreviations list to the end of the paper.

-       The results and discussion paragraph is very confusing to me. I suggest splitting the paragraph in two different paragraphs (results/discussion).

-       The figure legends are not very clear: I suggest specifying a little bit more what is reported in the figure. i.e in figure1A you have not even mention that the picture represents an immunoblot. Please, also add how many replicates you have done for each experiment.

-       Figure 1D, 2B and 2D: specify at which p value corresponds “*”

-       The figures are a bit blurred - can the resolution be improved?

-       Table 1: replace “*” with another symbol since it could be confused with the p value.

-       What does “pLF1, pLF3, pLF2 and pCTR” mean? You speak of peptides, but you need to be more precise.

-       In Material and methods, you talk about ACE2, HSPG, HSA, hLF, casein for the binding assay. Where were these molecules purchased from?

-       Which software have you used for the statistical analysis? Please add it.

-       Lines 156-160: I would move these sentences to a new paragraph “Conclusions”.

Author Response

Response to reviewer #4:

We thank the reviewer for the positive evaluation of our manuscript.

-       I suggest moving the abbreviations list to the end of the paper.

We moved the section [lines 279-284]. We thank for the suggestion.

-       The results and discussion paragraph is very confusing to me. I suggest splitting the paragraph in two different paragraphs (results/discussion).

We split the section. We thank for the suggestion.

-       The figure legends are not very clear: I suggest specifying a little bit more what is reported in the figure. i.e in figure1A you have not even mention that the picture represents an immunoblot. Please, also add how many replicates you have done for each experiment.

We expanded the figure legends as required.

-       Figure 1D, 2B and 2D: specify at which p value corresponds “*”

This information is now specified in the corresponding figure legends.

-       The figures are a bit blurred - can the resolution be improved?

For the evaluation, we used the state-of-the-art chemiluminescence image analyser Azure 280 (AzureBiosystems, Dublin, CA) and the AzureSpot software. The images were saved in the highest possible quality. For publication, we prepared the images in the Adobe Illustrator software and provide the highes-resolution TIFF formats. Some blurriness could have been caused by enlarging the images.

-       Table 1: replace “*” with another symbol since it could be confused with the p value.

We thank the reviewer for this observation. For better clarity, in the revised form we exchanged the symbols.

-       What does “pLF1, pLF3, pLF2 and pCTR” mean? You speak of peptides, but you need to be more precise.

Those abbreviation indeed mean the peptides. For better clarity, the abbreviations are now involved in the abbreviation list. We thank for this notion.

-       In Material and methods, you talk about ACE2, HSPG, HSA, hLF, casein for the binding assay. Where were these molecules purchased from?

The products were purchased from Sigma-Aldrich, as was described in the Materials section [line 213-214].

-       Which software have you used for the statistical analysis? Please add it.

We thank for this important notion, we used the Prism 10, as it is now listed accordingly [lines 266].

-       Lines 156-160: I would move these sentences to a new paragraph “Conclusions”.

We moved the paragraph as recommended [lines 163-167]. We thank for the suggestion.

Round 2

Reviewer 2 Report

Comments and Suggestions for Authors

Thank you very much for revising the manuscript. 

Author Response

Response to reviewer #2:

We thank the reviewer for positive evaluation.

Reviewer 4 Report

Comments and Suggestions for Authors

The authors have responded accurately and comprehensively to the comments I raised, making the work clearer and more appealing to readers. The only point that remains to be improved are the figures (in particular Figure 1) which are of low resolution and stretched.

I suggest to accept the paper after this minor revision.

Author Response

Response to reviewer #4:

The authors have responded accurately and comprehensively to the comments I raised, making the work clearer and more appealing to readers. The only point that remains to be improved are the figures (in particular Figure 1) which are of low resolution and stretched. I suggest to accept the paper after this minor revision.

As mentioned in our previous response, for the measurement of chemiluminescent signals from the blots, we used the image analyser Azure 280 (AzureBiosystems, Dublin, CA) and the AzureSpot software. The images were saved in the highest available resolution. For publication, we prepared the images in the Adobe Illustrator software and provide the highest available resolution TIFF formats. For the clarity and easier full and informative description we enlarged the images. Now, in the revised version we reduced sizes of the blot images as required and tried to keep in parallel the clarity of description. We keep the size of the blot in Figure 3 since it shows the two-dimensional BN-PAGE, which requires the good discrimination of the band separations. We thank the reviewer for the positive evaluation.
